# Drought drives rapid shifts in tropical rainforest soil biogeochemistry and greenhouse gas emissions

Christine S. O'Connell [1], Leilei Ruan[1] & Whendee L. Silver [1]

Climate change models predict more frequent and severe droughts in the humid tropics. How drought will impact tropical forest carbon and greenhouse gas dynamics is poorly understood. Here we report the effects of the severe 2015 Caribbean drought on soil moisture, oxygen, phosphorus (P), and greenhouse gas emissions in a humid tropical forest in Puerto Rico. Drought significantly decreases inorganic P concentrations, an element commonly limiting to net primary productivity in tropical forests, and significantly increases organic P. High-frequency greenhouse gas measurements show varied impacts across topography. Soil carbon dioxide emissions increase by 60% on slopes and 163% in valleys. Methane ($CH_4$) consumption increases significantly during drought, but high $CH_4$ fluxes post-drought offset this sink after 7 weeks. The rapid response and slow recovery to drought suggest tropical forest biogeochemistry is more sensitive to climate change than previously believed, with potentially large direct and indirect consequences for regional and global carbon cycles.

[1] Department of Environmental Science, Policy and Management, University of California, Berkeley, CA 94720, USA. Correspondence and requests for materials should be addressed to C.S.O'C. (email: coconn@berkeley.edu)

The near-constant warm temperatures and moist soils typical of humid tropical forests support high rates of carbon (C) cycling, and globally significant exchanges of C with the atmosphere[1,2]. Humid tropical forests have among the highest rates of soil carbon dioxide ($CO_2$) emissions globally[3] and tropical soils are an important source of $CH_4$[4]. These high rates of C cycling occur under low and fluctuating redox conditions[5–7], indicated here by soil oxygen ($O_2$) concentrations[6,7], which can increase heterotrophic respiration[6,8,9], $CH_4$ emissions[6,10], and soil P availability, a key nutrient that is frequently limiting to net primary productivity (NPP) in humid tropical ecosystems[11–14].

Climate models predict increasing drought frequency and severity in the humid tropics as a result of climate change[15–17], with poorly understood effects on biogeochemistry and the C cycle[18–24]. Data from throughfall removal experiments in moist and humid tropical forests show both increased[25] and decreased[26,27] soil $CO_2$ emissions, and both increased soil $CH_4$ emissions[28] and soil $CH_4$ consumption[26]. Few studies of greenhouse gas (GHG) emissions and biogeochemical cycling have captured natural drought events, and none have had the spatial and temporal resolution to determine patterns across complex ecosystems like humid tropical forests[29,30]. To better determine the effects of drought in tropical forests, high-frequency measurements located across key ecosystem units such as topographic zones are needed to capture both the spatial and temporal dynamics of these events[18]. Spatially distributed, continuous measurements such as these are critical to improve our understanding of how tropical forest functioning is likely to change under future climatic regimes.

In 2015, the Caribbean and South American tropics experienced an historic drought during the El Niño Southern Oscillation (ENSO) event[31]. We used a spatially- and temporally-rich continuous automated sensor network, in place before the onset of the drought, to determine its effects on tropical forest biogeochemistry and feedbacks to climate change. This unique opportunity allowed us to address the following questions: How does drought affect feedbacks to climate change through C GHG emissions and their associated drivers? How quickly do soil moisture, redox, GHGs, and soil P pools respond to drought and how fast do they recover? What are the spatial patterns in drought response and recovery? We used high-frequency measurements located across replicate ridge-slope-valley catenas in a humid tropical forest to test the hypothesis that soil drying increases aerobic respiration and $CH_4$ uptake due to higher $O_2$ availability. We hypothesized that valleys would be the slowest to respond biogeochemically to drought and the fastest to recover, due to high background moisture and low redox conditions. Finally, we hypothesized that drought would result in lower inorganic soil P availability due to the propensity of iron (Fe) oxides to bind P in well-aerated, highly weathered tropical forest soils.

## Results

**Rainfall patterns and drought intensity.** At El Verde Research Station, Puerto Rico (Methods), total precipitation in 2015 was only 2035 mm, compared to a mean and standard error of 4219 ± 772 mm per year for the previous decade (2004–2013), representing a 48% decline (Fig. 1, Fig. 2a, Supplementary Fig. 1). Drought conditions were distributed across the year: during 2015, 291 days had a negative rainfall anomaly in comparison to 2004–2013 rainfall trends (Fig. 2a). We found that there were four statistically distinct soil moisture regimes ($p < 0.001$): a pre-drought period (through late April), an acute drought period (April–August), a drought recovery period (August–November), and a post-drought period (ongoing). These drought periods were defined using structural change modeling (Methods; R package 'strucchange').

**Moisture and oxygen impacts during drought.** The drought resulted in rapid and significant moisture declines in soils that are typically moist year round[32]. Soil moisture, temperature, and $O_2$ measurements were made hourly (0–15 cm depth) from November 2014 to February 2016 using automated sensors across topographic gradients (Methods). Seven sensors of each type were installed in an automated soil sensor array along five ridge to valley transects for a total of 105 sensors (Supplementary Fig. 2). The sensor array was located within a 50 by 50 m area with a slope angle of 25° and vertical height of 5 m; sensor transects were placed at least 2.5 m apart.

Ridges, which had the driest pre-drought soils, saw the largest proportional declines in soil moisture, while the valley and lower slope soils experienced the largest percentage increase in soil $O_2$ availability. Soil moisture on ridges decreased from a mean of 36

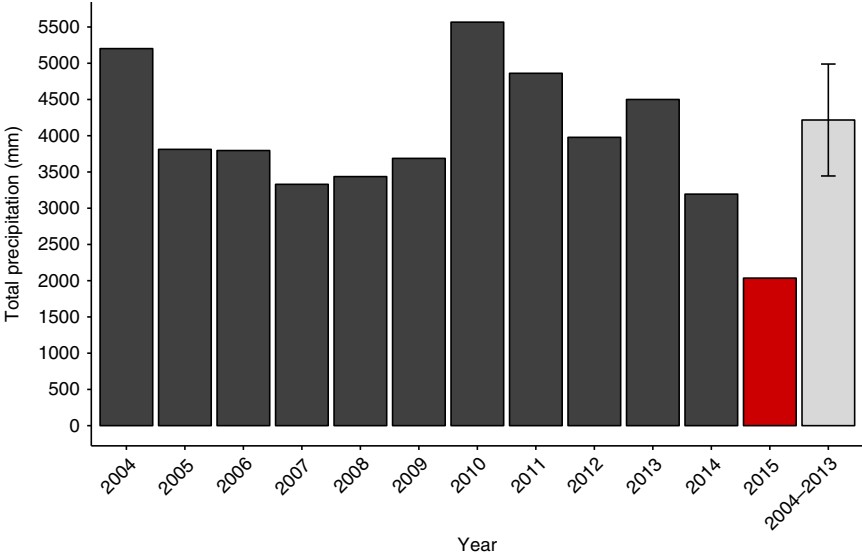

**Fig. 1** Recent annual precipitation data for El Verde Research Station. Years 2004–2013 serve as recent comparison (mean and ±1 s.d. reported); this study includes observational data collected from 2014 to 2015

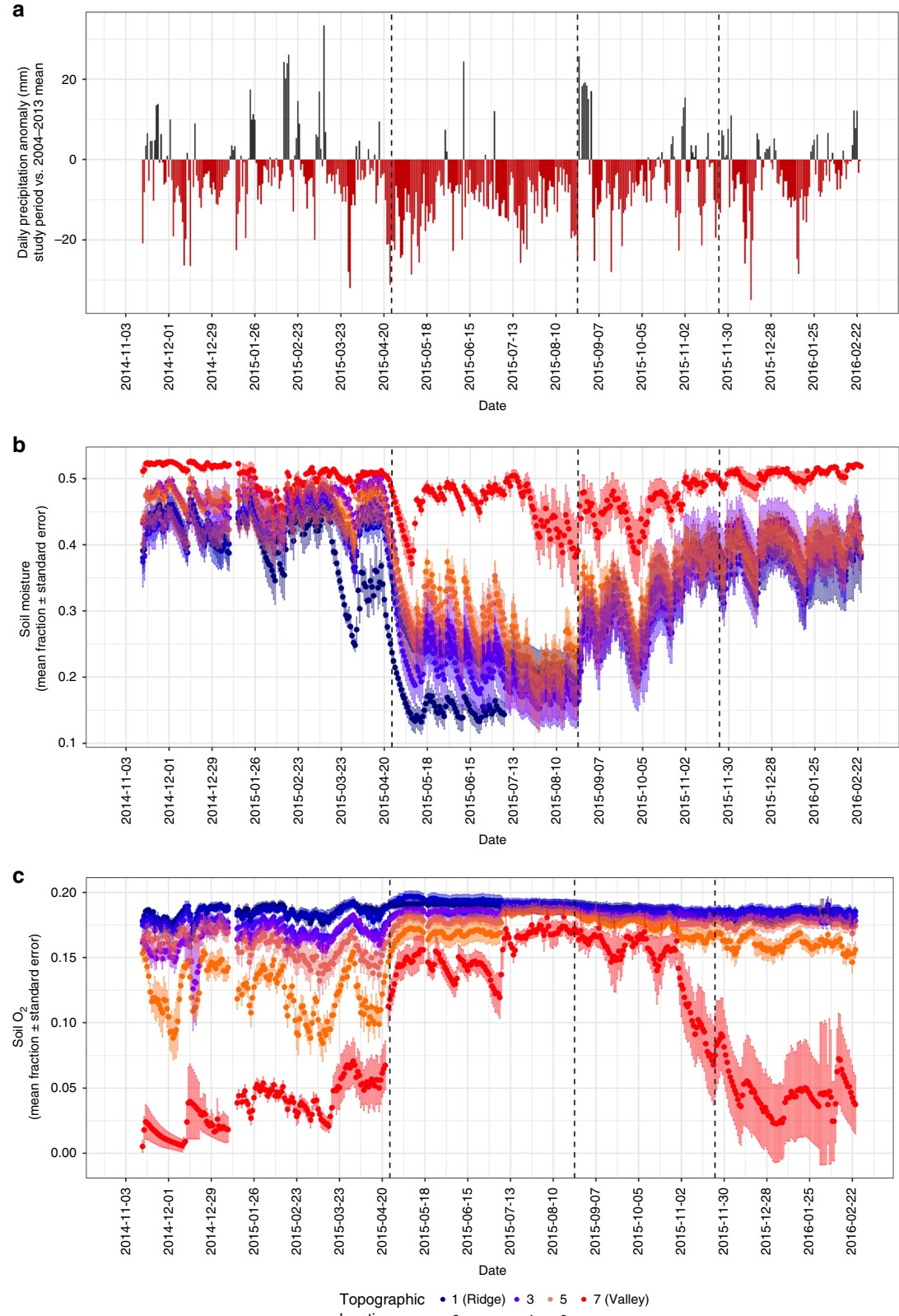

**Fig. 2** Time series of rainfall and soil abiotic variables over the study period. Panels show **a** daily precipitation anomaly (2015 vs. 2004–2013), **b** soil moisture fraction, and **c** soil oxygen fraction from November 2014 to February 2016. Soil moisture and oxygen data points are the daily mean for each of the seven topographic locations (ridge to valley); error bars represent ±1 s.e.m.

± 8% to 13 ± 2% over 21 days, representing a 63% decline (Fig. 2b). Valley bottoms, the wettest zone in the landscape, declined from a mean of 51 ± 1% to 37 ± 7% over the same period (Fig. 2b). Soil $O_2$ concentrations in the valley bottoms doubled in one week from a mean of 5.6 ± 4.1% to 11.2 ± 3.4%, and increased by 36% within the first month of the drought (Fig. 2c). These zones are typically hotspots of low redox processes such as methanogenesis and denitrification[6].

**$CO_2$ and $CH_4$ impacts during drought.** Drought significantly increased soil $CO_2$ emissions across all topographic zones ($p <$ 0.001, Fig. 3). Carbon dioxide emissions were greatest from slopes and lowest from ridges ($p < 0.001$). On slopes, soil respiration almost doubled from 3.79 ± 2.92 µmol m$^{-2}$ s$^{-1}$ prior to the drought to 6.06 ± 4.26 µmol m$^{-2}$ s$^{-1}$ during the drought period (Fig. 3). Drought led to a steep decline in valley emissions of $CH_4$, from 17.43 ± 29.60 nmol m$^{-2}$ s$^{-1}$ prior to the drought to 1.67 ± 4.09 nmol m$^{-2}$ s$^{-1}$ during the drought period, while also increasing the sink strength of $CH_4$ in the ridge and slope sites— the ridge sink increased by 76% during drought while slopes went from being slight sources of $CH_4$ to consuming 1.93 ± 1.73 nmol m$^{-2}$ s$^{-1}$; $CH_4$ emissions during the pre-drought and drought periods were significantly different ($p < 0.05$, Fig. 3, Supplementary Fig. 3). Trace gas measurements were taken using nine automated gas flux chambers, three in each topographic zone (ridge, slope, and valley, Supplementary Fig. 2). A Cavity Ring-Down Spectroscopy (CRDS) gas analyzer was used to measure fluxes of $CO_2$ and $CH_4$. The dataset presented here includes over 6300 $CO_2$ and $CH_4$ flux observations over 10 months, providing a uniquely robust estimate of how drought impacts trace gases (Supplementary Fig. 3, Methods).

**Soil P and other chemical properties during drought.** Drought drove significant declines in plant-available, inorganic forms of P across the ecosystem ($p < 0.001$, 0–15 cm depth, Fig. 4a). Inorganic P concentrations were significantly higher in valley bottoms than in upper topographic zones both pre- and post-drought ($p < 0.001$), with valleys experiencing the largest proportional decline, a 60% loss in inorganic P concentrations as a result of the drought. This decline may have been driven by Fe–P bonding which can increase after Fe oxidation[33]. The decrease in inorganic P pools was concurrent with a significant increase in oxidized iron (Fe(III)) and a significant decline in reduced iron (Fe(II)) as soils became more aerated ($p < 0.01$, 0–15 cm depth, Fig. 4c, d). Soil pH also dropped significantly in valley soils post-drought ($p < 0.001$, 0–15 cm depth, Fig. 4e), another indication of a change from a reducing to an oxidizing environment in these Fe-rich soils[34].

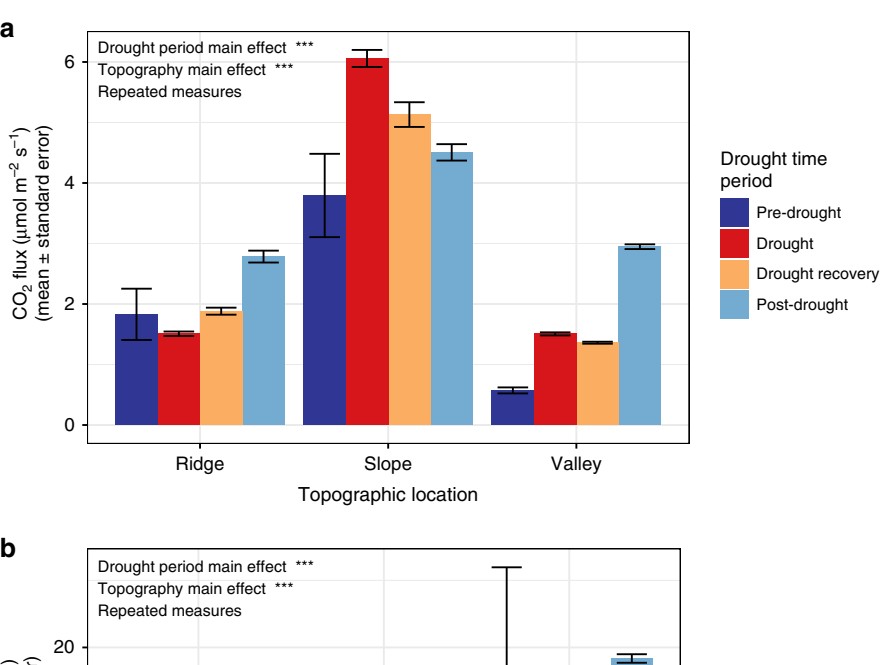

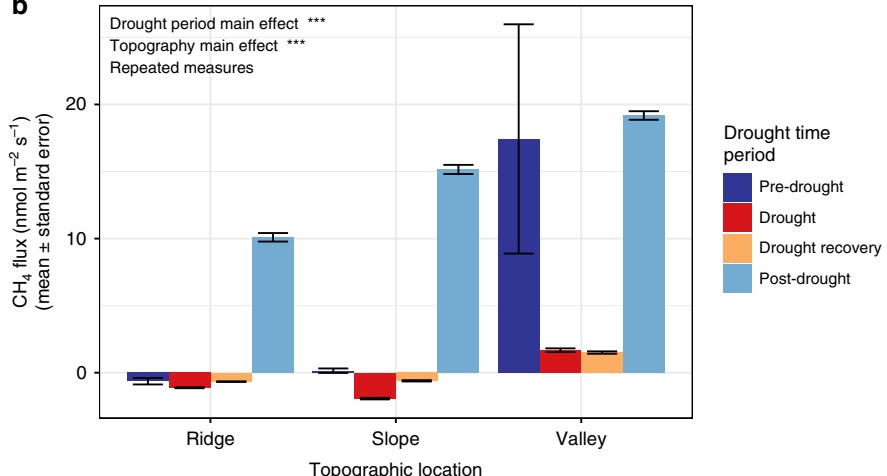

**Fig. 3** Soil greenhouse gas emissions across topographic zones. Bars are the mean flux of $CO_2$ (**a**) and $CH_4$ (**b**) for each drought period; error bars represent ±1 s.e.m. Statistical results are reported from a two-way, repeated measures ANOVA in which drought period and topographic location were predictors of GHG fluxes; effects are marked as *** when *p*-value <0.001, ** when *p*-value <0.01, * when *p*-value <0.05, and as NS when *p*-value ≥0.05

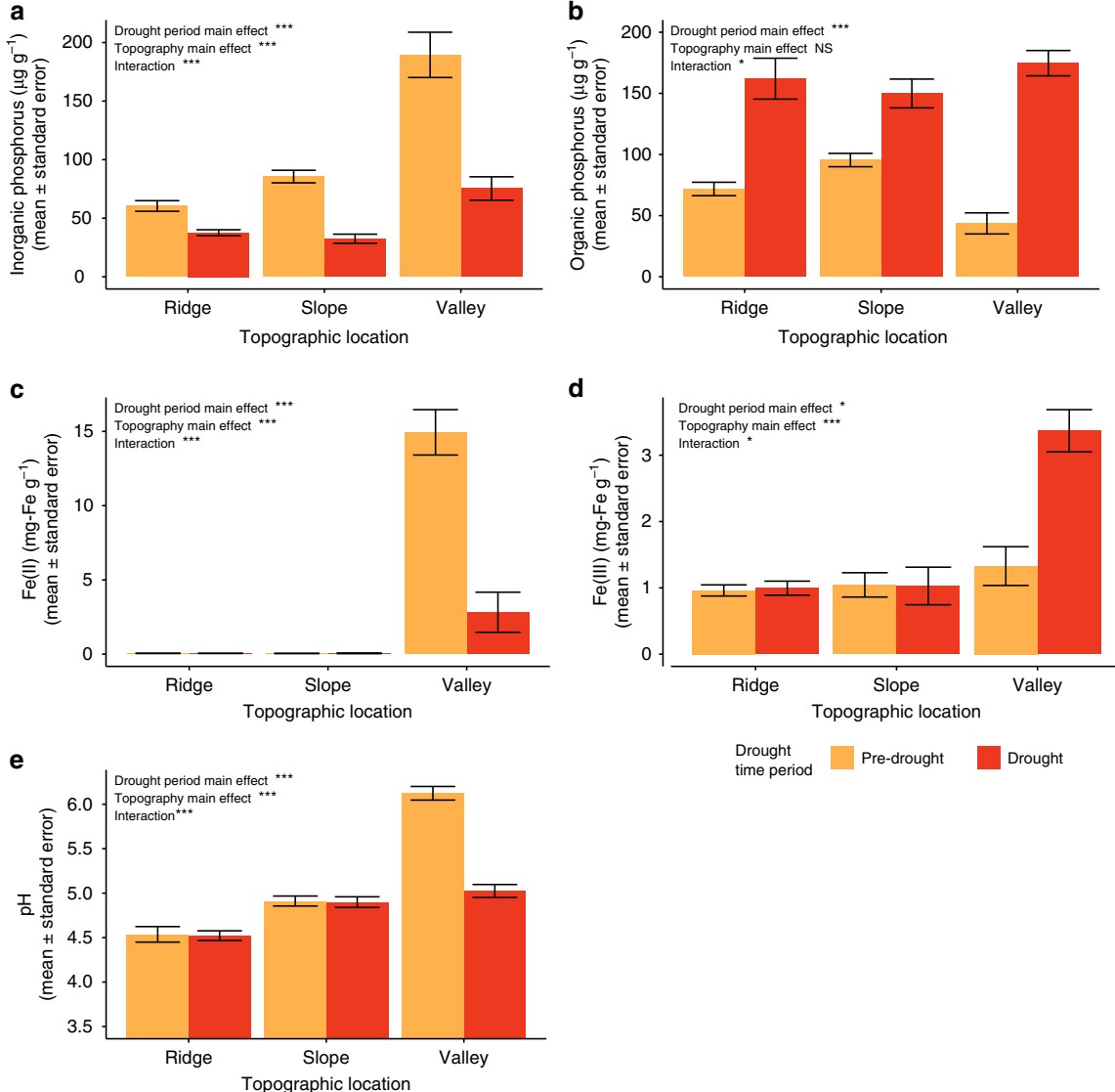

**Fig. 4** Soil biogeochemical variables before and during drought. Panels show **a** inorganic P, **b** organic P, **c** Fe(II), **d** Fe(III), and **e** pH. Bars are the mean value; error bars represent ±1 s.e.m. Statistical results are reported from a two-way ANOVA in which drought period, topographic location, and their interaction were predictors of the given soil variable; effects are marked as *** when *p*-value <0.001, ** when *p*-value <0.01, * when *p*-value <0.05, and as NS when *p*-value ≥0.05

Organic P concentrations were not significantly different across topographic zones (0–15 cm depth, Fig. 4b), and increased significantly with drought ($p < 0.001$), particularly in valley bottom soils, where the mean organic P concentration increased by 301% (increase when accounting for standard errors = +215% to +428%). Organic P concentrations in ridge and slope soils rose by 126% (+88 to +170%) and 56% (+37 to +80%), respectively. Organic P, although not immediately available for plant uptake, may provide an important reservoir of P during post-drought conditions.

**Moisture and oxygen recovery after drought**. The drought recovery period (Fig. 2) was ~65% of the length of the drought period with regard to soil moisture, imposing a much longer drought condition belowground than observed in the precipitation record alone. Valley soils recovered soil moisture more slowly than ridge soils (0.06% vs. 0.15% moisture recovery day$^{-1}$) (Fig. 2b). Post-drought soil moisture was not significantly different from the pre-drought period (Supplementary Data 1). Soil

$O_2$ had yet to recover for the two lowest slope positions after the drought recovery period (Fig. 2). An incomplete recovery of moisture-$O_2$ dynamics was likely driven by changes in soil structure after rapid soil drainage and the lengthy period of drying.

**$CO_2$ and $CH_4$ recovery after drought**. The drought, drought recovery, and post-drought periods all had significantly higher $CO_2$ fluxes than the pre-drought period ($p < 0.001$, $p < 0.01$, $p < 0.001$ respectively; Fig. 3). $CO_2$ emissions were slow to recover to pre-drought levels, perhaps due to the slow decline in soil $O_2$ concentration post-drought, which effectively continued the belowground drought conditions. However, the post-drought period had significantly higher $CH_4$ emissions than pre-drought, drought, and drought recovery periods ($p < 0.001$, Fig. 3). During the post-drought period, valley $CH_4$ emissions recovered to within 10% of pre-drought emissions, and ridge and slope emissions were persistently above their pre-drought levels. Ridge soils had net consumption of 0.64 ± 0.76 nmol m$^{-2}$ s$^{-1}$ pre-

drought and emitted $10.12 \pm 7.84$ nmol m$^{-2}$ s$^{-1}$ post-drought. Slope soils increased their pre-drought emissions of $0.15 \pm 0.69$ nmol m$^{-2}$ s$^{-1}$ by a hundredfold to $15.16 \pm 7.03$ nmol m$^{-2}$ s$^{-1}$ post-drought.

**Ecosystem scale estimate of GHG emissions**. During the drought and drought recovery period, there was a marked drop in hot moments of CH$_4$ fluxes across the ecosystem (Supplementary Fig. 3). We used the highest 10% of fluxes (90th percentile) to identify hot moments across the datasets. The 90th percentile of pre- and post-drought emissions of CH$_4$ were an order of magnitude higher than the 90th percentile flux rate during the drought and drought recovery periods (90th percentile rates of 11.77, 1.88, 1.66, 22.01 nmol CH$_4$ m$^{-2}$ s$^{-1}$ during each consecutive drought period). During the drought and drought recovery periods, 70 kg CO$_2$e ha$^{-1}$ (68–70 kg CO$_2$e ha$^{-1}$ 95% CI) was consumed as CH$_4$ at the ecosystem scale. The dramatic increase in CH$_4$ emissions during the first 50 days of the post-drought period offset 99% (93–102% 95% CI) of the CH$_4$ sink during the drought. We used the distribution of ridges (17%), slopes (65%), and valleys (18%) as a first approximation of fluxes at the ecosystem scale and converted CH$_4$ fluxes to kg CO$_2$e emitted per hectare using the 100-year global warming potential value[35] (e.g., 34 for CH$_4$) (Methods, Supplementary Fig. 4). During the same time period, CO$_2$ emissions rates increased above pre-drought levels, and led to an additional 12.05 Mg CO$_2$e ha$^{-1}$ (11.26–12.95 Mg CO$_2$e ha$^{-1}$ 95% CI) in emissions.

## Discussion

Across the observed topographic gradient, the effects of drought belowground occurred suddenly, with large changes in moisture and O$_2$ concentrations over a 1- to 3-week time scale, and resolved slowly (over 12 weeks), extending the effective drought period. These results show that the ecological impacts of drought cannot be determined from the rainfall record alone. Results also emphasize the need to incorporate topography and other key drivers of biogeochemical variation into investigations of global change impacts on forest ecosystems. For example, responses of moisture and soil O$_2$ concentrations varied dramatically, but consistently, across the replicate hillslope transects. Incorporating natural variation facilitates upscaling to better determine the impacts of drought at the larger, whole-ecosystem scale.

The system-wide 2015 drought led to increased soil CO$_2$ fluxes and a further increase in fluxes post-drought. Higher CO$_2$ fluxes could result from increased heterotrophic and/or autotrophic respiration, and determining the relative proportions is notoriously difficult in natural ecosystems[36]. Higher root respiration[37] in drought-stressed plants[29,38] could drive higher autotrophic respiration rates. Drought experiments have observed increases in tree[39–41] and root[37,42] mortality with drought, with potential short- and long-term impacts on CO$_2$ fluxes[21]. The Amazon-wide 2010 drought led to decreases in root respiration rates as trees shifted primary production from fine root to canopy leaf tissues[43]. Direct measurements of plant C allocation are not available in this system from the drought period, but indirect evidence suggests shifts in plant strategies. Litter traps during the drought period showed an increase in leaf litter inputs at the onset of drought, but no increase in total cumulative leaf input over the course of the study period (A. Hogan and J. Zimmerman, personal communication). A large initial leaf drop event could presage a shift in NPP towards canopy growth during post-drought periods. If this were the case, the observed increase in soil CO$_2$ emissions would result dominantly from heterotrophic respiration. Increased organic matter inputs during drought (i.e., due to root exudation) could also drive an increase in heterotrophic

respiration[44,45]. Heterotrophic respiration rates could have been stimulated by a flush of nutrients during the drought recovery and post-drought periods; prolonged dry periods can be associated with soil nutrient accumulation which is then made available during soil rewetting[46]. A short-term increase in soil nutrient availability could also stimulate autotrophic respiration rates.

An additional explanation of increased soil CO$_2$ emissions during drought could be that newly-dry soils warmed, which led to a temperature-driven change in emissions. While soil temperatures increased slightly during drought (Supplementary Fig. 5), they did so gradually, following a seasonal pattern that mirrored local air temperature measurements (Supplementary Fig. 5), in marked contrast from the sudden change in GHG emissions (Supplementary Fig. 3) and soil moisture patterns (Fig. 2). Soil temperatures also fell during the post-drought period, while CO$_2$ emissions remained higher than pre-drought flux rates in all topographic positions, indicating that soil temperature is insufficient to explain changes in CO$_2$ flux patterns. Across the study period, soil and air temperatures were positively correlated while soil temperatures did not correlate well with either CO$_2$ or CH$_4$ flux rates (Supplementary Fig. 6), suggesting that temperature changes are not sufficient to explain observed GHG patterns.

Upper topographic zones (slopes and ridges) consumed CH$_4$ prior to the drought and drought predictably increased the CH$_4$ sink strength, similar to results from throughfall exclusion experiments that also observed increases in soil CH$_4$ consumption[26,47,48]. Valleys were net emitters of CH$_4$ to the atmosphere throughout the study, although the drought significantly decreased emission rates. The dramatic increase in CH$_4$ emissions from all topographic zones following drought was a surprising result. This may have resulted from increased organic C availability together with re-saturation and creation of anaerobic microsites.

The large increase in soil organic P after drought onset was also an unexpected result. There are several potential mechanisms that may have contributed to the patterns observed. Organic P increases could potentially be explained by a pulse of inputs associated with plants dropping their leaves without resorption of leaf nutrients due to drought stress[25,37], consistent with the aforementioned observed increase in leaf litter inputs at the beginning of the drought period. Changes to microbial activity may have also contributed to organic P increases. Drought conditions could have led to a slowdown of extracellular phosphatase activity, an effect that has been observed in other systems[49,50]. This pattern could reconcile higher CO$_2$ emissions with higher organic P concentrations, as microbial decomposition could proceed without liberating the P in soil organic matter. Alternatively, rapid and considerable microbial uptake of the newly available inorganic P would have been measured as organic P. Tropical drought has previously been shown to lead to lower decomposition rates[51,52], which could lead to a subsequent accumulation of organic P. Given that we observed an increase in soil CO$_2$ emissions during drought (Fig. 3a), any substantial decrease in decomposition rate would have had to have been offset by a large increase in root respiration, making this explanation for organic P increase less likely.

Results of the 2015 drought showed that humid tropical forests are very sensitive to drying events. Drought resulted in an increase in soil respiration that persisted during the drought recovery and post-drought periods. Drought led to a net reduction in CH$_4$ emissions, driven largely by the biogeochemical dynamics on slopes and in valleys. However, the dramatic increase in CH$_4$ emissions across the catena in the post-drought environment offset approximately 99% of this sink in a matter of weeks. Large and rapid shifts in soil aeration were likely drivers of

these dynamics. We also found that inorganic P, a key nutrient often limiting NPP in these ecosystems, declined by up to 60% during the drought. Organic P increased, and may provide an important reservoir for plants and microbes over time. Taken together, results show that drought has important implications for biogeochemistry at the ecosystem and global levels, via both direct effects (e.g., soil drying, changes in trace gas emissions) and indirect effects (e.g., declines in inorganic P availability, increases in organic P concentrations). These data are critical for reducing uncertainties surrounding how terrestrial C and nutrient cycles will be modified by climate changes[1,35,53].

## Methods

**Field location information**. Research was conducted in the Luquillo Experimental Forest (LEF), Puerto Rico, USA (Lat. 18°18′N; Long. 65°50′W). The forest is congruent with El Yunque National Forest managed by the US Forest Service. The LEF contains approximately 11,500 ha of contiguous forest area, spanning an elevation gradient from approximately 350 to 1075 m above sea level. The LEF has been well characterized geologically and ecologically as part of ongoing Long-Term Ecological Research and Critical Zone Observatory projects[6,26,54–56]. Soils in the LEF are derived from volcanoclastic sediments with quartz diorite intrusions[57].

El Verde Research Station, where this research took place, is located at ~350 m a.s.l. elevation. Mean monthly temperatures range from 20.6 °C to 25.8 °C with an annual mean temperature of 23.0 ± 1.9 °C (means derived from 1975 to 2004 temperature record)[58]. The forest can be classified as subtropical wet forest and the plant community is mature tabonuco (*D. excelsa* Vahl) forest[59]. Soils at the field site are clay-rich Ultisols (Supplementary Tables 1 and 2) derived from volcanoclastic parent material.

**Experimental design**. An automated sensor array was installed near the El Verde Research Station (Lat. 18°32′N; Long. 65°82′W). The array was composed of five replicate transects, each with seven topographic locations from ridge to valley; the sensor transects were located about 2.5 m apart over a 50 by 50 m plot on a slope of angle 25° over 5 m vertical distance (overall slope angle = 25°, steepest area (upper slope) angle = 55°; both measured using a standard field clinometer). The location was chosen to be generally representative of the vegetation, soils, and topographic variability of the larger watershed ecosystem[54].

Each of the 35 measurement points contained a sensor cluster with a galvanic $O_2$ sensor (Apogee Instruments, Logan, UT, USA) and a time-domain reflectometry sensor (Campbell Scientific, Logan, UT, USA), both of which were installed in the top 15 cm of soil (Supplementary Fig. 2). The moisture, temperature, and $O_2$ sensor clusters were installed at a random location within each of the seven zones along the catena; sensors were within approximately 5–10 cm of each other as root and rock placement allowed. Nine automated gas flux chambers (Eosense, Nova Scotia, Canada) were installed within the sensor array; three chambers each were distributed randomly within ridge, slope, or valley zones (Supplementary Fig. 2). The chamber sites corresponded to the relatively flat ridge top, the mid slope, and the valley bottom of the sensor array transects. Continuous sensor measurements began in November 2014. Trace gas measurements began in February 2015.

**Soil sensor and rainfall measurements**. Daily mean soil $O_2$ and moisture measurements were compared with precipitation patterns. Soil $O_2$ sensors were installed in the top 15 cm of soil in gas-permeable soil equilibration chambers (295 mL, 5 cm diameter, 15 cm height) (*sensu* Liptzin et al. 2011[55]) at each of the 35 sensor locations in the topographic array. Data from these sensors were collected hourly using data loggers (Campbell Scientific, Logan, UT, USA) and multiplexers (Campbell Scientific, Logan, UT, USA). Precipitation data were collected at a nearby rain gauge located at El Verde Research Station, approximately 500 m from the field site. This gauge is administered by the Luquillo LTER as part of the long-term ongoing climate monitoring program in the LEF. Rainfall has been recorded daily or semi-daily since 1964 and this record was used to report precipitation during the study period as well as historical precipitation patterns (historical data catalogued on the Luquillo LTER datanet[60]).

**Gas flux measurements**. To determine patterns in trace gas fluxes across the soil atmosphere interface we used nine automated surface flux chambers deployed with the sensor network plots (3 ridge, 3 slope, 3 valley; Supplementary Fig. 2, Supplementary Table 3, Supplementary Data 2). Automated flux chambers were connected to a multiplexer, which dynamically signaled chamber deployment and routed gases to a Cavity Ring-Down Spectroscopy (CRDS) gas analyzer (Picarro, Santa Clara, CA, USA)[61]. The automated chambers, multiplexer, and CRDS gas analyzer were powered by a generator (Honda, Tokyo, Japan) with the generator, multiplexer, and CRDS gas analyzer housed in a shed away from the array to avoid contamination from generator exhaust. When chambers were not measuring a flux, lids were not in contact with the chamber base, and instead were held

approximately 10 cm above and 5 cm outside of the chamber base circumference, in order to minimize impact on the sampling area and ensure that precipitation would reach soil within the chamber footprint. Chambers were closed for a 10-min sampling period with a 3-min flushing period between chamber measurements.

Trace gases were measured from one chamber at a time; a full cycle through the nine chambers took approximately 2 h and occurred continuously leading to a maximum of 12 measurements per chamber per day. Sampling occurred daily unless instrument malfunction prevented sampling, which occurred because of instrument failure or debris (branches or large leaves) inhibiting chamber closure. The array was inspected at least twice each day to minimize these events. Instrument-related gaps in the data record are associated with instances in which the multiplexer, CRDS gas analyzer, or generator needed repairs (Supplementary Fig. 3). Automated chambers and the CRDS gas analyzer also recorded chamber and instrument temperatures, relative humidity values, and pressure during the flux measurements. Fluxes recorded during periods of high values of chamber and instrument temperatures, relative humidity values, and pressure were removed from the data record. After data cleaning and accounting for days in which data could not be collected, 6479 $CO_2$ flux observations and 6379 $CH_4$ flux observations remained from 150 unique days (out of 326 possible sampling days).

Fluxes of $CO_2$ and $CH_4$ were calculated using software developed to work in tandem with an automated chamber-CRDS gas analyzer set up (Eosense EosAnalyze-AC v. 3.4.2). For each measurement, two flux rates were calculated, one using a linear model and one using an exponential model *sensu* Creelman et al.[62].

Dataset quality assessment and control were subsequently performed in R (R v. 3.2.2). Fluxes were removed from the final dataset if they were associated with anomalous temperature, moisture, or pressure readings, if the initial concentrations of $CO_2$ or $CH_4$ were substantially higher or lower than ambient values (potentially indicative of a malfunctioning flush period) or if the chamber deployment period was less than 9 min or more than 11 min. The choice between the linear and exponential flux rate models was decided upon using the estimate uncertainty to estimate ratios, and in cases where both the linear and exponential models produced high uncertainty, the flux was eliminated from the dataset. Detailed results of all GHG fluxes are provided in Supplementary Tables 3 and 5.

**Soil variable sampling and processing**. We sampled soils from ridge, slope, and valley locations within the catena from 0 to 15 cm depth before the onset of the drought (April 2015) and during the height of the drought (July 2015). Six replicate samples were taken from each topographic zone at each sampling time point. Soils were transported to Berkeley, CA, USA and processed within 5 days of collection for pH and concentrations of Fe(II), Fe(III), organic P, and inorganic P.

We performed 0.5 mol $L^{-1}$ HCl extractions on 5 g of wet soil, which were analyzed for concentrations of total Fe (i.e., Fe(II) + Fe(III)) and Fe(II) on a spectrophotometer (Thermo Scientific Genesys 20, Fisher Thermo Scientific) (*sensu* Liptzin and Silver 2009 ref.[63]). Fe(II) concentrations were measured directly while Fe(III) concentrations were calculated as the difference between total Fe and Fe(II) concentrations. Soil pH was measured on samples of 1.5 g of wet soil in DI water using a pH probe (Denver Instrument Ultrabasic pH/mv Meter (UB-10)).

We performed a Hedley phosphate extraction with two extraction steps[64]. We first extracted 1.5 g of wet soil in 0.5 mol $L^{-1}$ sodium bicarbonate to measure the concentration of organic phosphorus. We followed this extraction with a 0.1 mol $L^{-1}$ sodium hydroxide extraction to measure the concentration of inorganic phosphorus. The extract was measured on a spectrophotometer (Thermo Scientific Genesys 20, Fisher Thermo Scientific).

Several one-time soil variables were measured, in all cases with four replicate samples taken for each topographic location (ridge, slope, valley). Soil bulk density was measured in August 2016 using standard volume cores (height 10 cm, diameter 6 cm) that were pounded into the soil, surrounded by an outer core to prevent soil compaction. Samples were oven dried at 105 °C for 72 h and then weighed. Air dried soil was analyzed for percent C and percent N using an elemental analyzer (CE Elantec, Lakewood, NJ) in December 2016. Soil texture was analyzed using the hydrometer method[65] in April 2015. Detailed results of all soil variables are provided in Supplementary Table 4.

**Statistical analyses**. Structural change modeling (also referred to as piecewise regression) was performed to determine how many distinct 'segments' there were in the soil moisture record (specifically, whether there were piecewise changes in the linear relationship between soil moisture and time). We used those segments to define the drought periods. The R package 'strucchange' was used to define the optimal number of partitions in the soil moisture data[66]. Splitting the data into four segments with three breakpoints had the lowest BIC value (in comparison to using 1, 2, or 4 breakpoints) with those breakpoints placed at time point 1099, 1942, and 2588 of the dataset (corresponding to 2015-04-25, 2015-08-24, and 2015-11-24). We performed a generic structural change test on a time series of F-statistics computed across the study period to calculate the overall p-value of the chosen segment set[67]; $p < 0.001$.

Soil Fe and P concentrations and soil pH data were analyzed using a two-way analysis of variance (ANOVA) where each response variable (e.g., concentration of organic P) was modeled using drought period and topographic location as predictor variables (model details in Supplementary Data 1). A repeated measures

ANOVA was not performed because this was not a repeated measure (soils sampled twice from randomly selected locations within topographic zones). Drought period was explicitly considered in the model. For significant models, a post-hoc Tukey's HSD test was performed. Bulk density, soil texture, percent C, and percent N were analyzed using a one-way ANOVA with subsequent Tukey's HSD test when relevant to determine whether topographic location was a significant predictor of each response variable (Supplementary Tables 1 and 2).

Soil–atmosphere $CO_2$ and $CH_4$ flux data were analyzed using a repeated measures two-way ANOVA with subsequent post-hoc Tukey's HSD test to determine whether drought period or topographic location explained soil gas flux rates (model details in Supplementary Data 1). In this case, a repeated measures ANOVA was considered appropriate because there were many repeated measurements taken from the same chambers within each drought time period. ANOVA diagnostic figures are found in Supplementary Figs. 7 and 8. The $CO_2$ flux data were log transformed prior to performing the ANOVA based on visual inspection of Q-Q and Trellis diagnostic plots. In all cases, models were deemed statistically significant at $p < 0.05$. All statistical analyses were performed in R (R v. 3.2.2). Unless otherwise noted, data were not transformed prior to statistical analyses.

We used the 100-year global warming potential values[35] of $CH_4$ (34) and $CO_2$ (1) to convert t $CO_2$ emitted per hectare and t $CH_4$ emitted per hectare into t $CO_2$e emitted per hectare. To calculate the total emissions over the study period from a general forest hectare in this system, we calculated a weighted average of the mean flux rate during each drought time period from each topographic zone, and weighted those averages by the proportion of the ecosystem that topographic zone represents (ridge = 17%, slope = 65%, valley = 18%, per Scatena and Lugo 1995 ref. [68]). We then ran simulations in R (R v. 3.2.2) at a daily time step to estimate the cumulative emissions over the 326-day study period: one simulation set presumed that the mean and standard deviation of the pre-drought flux rates for $CO_2$ and $CH_4$ continued over the time period (a baseline scenario), while a second simulation set used the mean (±s.d.) flux rate during the drought period, drought recovery period, and post-drought period when estimating the cumulative emissions. Each cumulative emissions estimate is the mean and 95% confidence interval of 5000 Monte Carlo simulations. Detailed results of all statistical analyses are provided in Supplementary Tables 3, 4 and Supplementary Data 1 and 2.

**Data availability**. Comprehensive datasets are available via the Luquillo LTER online data portal (http://luq.lter.network/datacatalog), including all gas flux, soil biogeochemical and soil abiotic variable data (dataset 'short name' ID 'LUQMetadata199', keywords 'greenhouse gas array').

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

## Acknowledgements

The authors thank R. Salladay, H. Dang, N. Nickerson, and W. Sereda-Meichel for their assistance in the field; N. Nickerson, C. Creelman, W. Sereda-Meichel, and S. Yasuhara for instrumentation technical support; and H. Dang, T. Anthony, S. Ahmed, and O. Gutiérrez del Arroyo for assistance in the lab. El Verde Research Station provided logistical support. This work was supported by grants from the Department of Energy (TES-DE-FOA-0000749) and National Science Foundation (DEB-1457805) to W.L.S. as well as the NSF Luquillo Critical Zone Observatory (EAR-0722476) to the University of New Hampshire and the NSF Luquillo LTER (DEB-0620910) to the University of Puerto Rico. W.L.S. received additional support from the USDA National Institute of Food and Agriculture, McIntire Stennis project CA- B-ECO-7673-MS.

## Author contributions

W.L.S. conceived the study. W.L.S. and L.R. designed the experiment. L.R. and C.S.O. collected the data. W.L.S. and C.S.O. analyzed the data and drafted the paper. L.R. helped with result interpretation and writing.

## Additional information

**Competing interests:** The authors declare no competing interests.

