## [Peer Review File · Nature Communications]

Reviewers' comments:

Reviewer #1 (Remarks to the Author):

Summary: This article describes the first example of "continuous-monitoring" automated chamber measurements of CO₂ and methane emissions from a tropical forest. Over one year, soil emissions were measured over a total of 150 unique days at three topographic positions in a lowland tropical rainforest. The study period encompassed a significant drought, enabling description of pre-during- and post-drought emissions, which varied. Soil O₂ and H₂O contents were simultaneously monitored through time in the same plot, as were soil inorganic and available P contents.

Temperature data are notably absent. Data are presented in figures only, minimizing their usefulness to others. The main findings are (I) drought increases CO₂ emissions but decreases CH₄ emissions (Fig. S5), possibly via O₂, and (II) the responses of both CO₂ and CH₄ emissions through time varied among topographic positions (Fig. 3). The lack of temperature data precludes the conclusion that dry soils, which cause soils to warm, caused the observed changes in gas fluxes.

Line:

"key nutrients" includes only two forms of P, organic and inorganic, or are you including Fe redox state as a nutrient? Would "soil phosphorus" be more appropriate?

It is important that the reader understands what "landscape-scale" means. According to the article title, it means a tropical rainforest. On the other hand, a standard hectare is referred to in the methods. In my experience, no single hillslope accurately reflects a tropical rainforest. I recommend that you avoid the "landscape" concept and word. This was a case study of a single hillslope; treat it as such.

I suspect that "there are no continuous" underlies a perceived value of this research. The trace gas measurements reflect repeated measures from nine distinct chambers. How that sample size was magnified by segregating by position and time-period needs to be explained more clearly, for the reader to properly interpret the findings. As I read it, there should be 36 total numbers, with a sample size and error terms, which underlie all of bars in Figure 3. As those are that backbone of this work, I suggest that they be included in a table in the appendix.

51-56 There are no objectives, hypotheses or questions stated, and thus no purpose identified, for this article. As a result, there is no logical organization, and the reader is forced to simply read along and see what comes next. Compounding this difficulty, there are no subheadings or other clues that might provide organization, and guide the flow of logic for the reader. Then, the concluding and introductory paragraphs are nearly the same. Your interesting findings, if they exist, are well disguised within the disorganization.

82-84 The clear statements of findings are welcome. However, there is no reference to any interaction term, so the statistical model is not obvious. Compare the (overly tiny) statistical results shown on Figure 3 with those on Figure 4: they are different! This suggests selective statistics, and I am quite sure that you want to be certain to avoid such an appearance. Basically, as a result, none of the results that you present are clearly derived from discernible statistical tests. This problem is magnified by the complete lack of any information about degrees of freedom or sample sizes, that is, basic information that informs the reader that the data was properly analyzed and interpreted. It is not included in the methods or statistical methods either.

97 The abstract (line 26) highlights only that inorganic P declined, without mentioning the opposing and apparently equal increase in organic P, and this same bias occurs in the concluding paragraph, at line 178. This again appears to be selective presentation, which is misleading.

123 The interpretation of the soil moisture -- O₂ relationship via linear regression is not even a tiny bit convincing. For one thing, sample sizes strongly influence R² and slopes. For another, it is obviously not a linear relationship. The conclusion made seems unwarranted, and are poorly described (in terms of what it is and what change occurred). Delete

157 It is gratifying to see that you did not forget to mention hot spots and hot moments. But it is unfair to ask the reader to first find Figure S3, and then interpret it properly, and then come to the same conclusion that you did, i.e., "a marked drop". Figure S3 presents means and standard deviations only – no hot spots or moments. Also, it is not clear how hot spots emerged when there were only nine chambers in (apparently) fixed positions. Are hot spots and hot moments the same

things?

Figure S5 This figure would be far more clear if (a) it included the entire period of study, with four time periods and (b) used molar values on the vertical axis. The values used herein are very challenging but could be simple. Simply use CO₂-Ce (mol area⁻¹ time⁻¹). But state your correction for methane (e.g., 86%), which you do not, anywhere.

173-175 Thresholds and lags were also observed. Again, however, these are presented as results without clear justification, and free of any statistical basis or meaning.

210 Figure S2 lacks important information that also is not provided in the text methods: What is the elevation difference between the ridgetop and valley? What were distances between H₂O/O₂ transects? What was the overall plot size?

332 The data from the single hillside array are applied to the forest landscape as a whole, to deduce the "rainforest" response to drought. Is this warranted?

335 This statistical approach presumes that the observed fluxes were normally distributed and had similar variances among groups, including across time periods when moisture content varied. This is exceedingly unlikely. Normal statistics are unwarranted unless you document that variances among compared groups were not different.

Reviewer #2 (Remarks to the Author):

This is an interesting and solid study that adds some unique information to a growing understanding of drought in tropical forests. The substantial amount of data spanning the drought and post-drought period offers a real plus, and in general I think this MS is worth considering for publication in Nature Comms. I do have a couple comments I believe need further consideration:

- The organic P increases are quite large – surprisingly so. They seem too large to be explained by a difference in decomposition, and given the increase in CO₂ emissions invoking a decrease in decomp for the organic P change doesn't really make sense. The leaf flush perhaps makes more sense but even that seems a bit tenuous given the time dynamics. I don't have a great explanation for what truly drove this, but would like to see the authors think on this one some more. E.g. are you seeing a big increase in microbial uptake of inorganic P pools under drought and conversion of that Pi into organic pools? Not sure...
- One of the truly hard things to do in interpreting the drought response of CO₂ is to know how much is from heterotrophs vs tree dynamics (so I'm sympathetic). There is some speculation in the MS about reasons for the change, but not much in terms of any supporting information that might help point the way on what's most responsible. Do you have any direct evidence for changes in decomposition rates and/or changes in plant C allocation? E.g. note the Doughty et al Nature paper on drought response that focused on C allocation responses in trees, demonstrating how critical this can be to interpreting the true C balance response of the system.
- Likewise, the post-drought further increase in CO₂ could be coming from multiple sources, again would be good to know if there are any supporting data to help interpret. Also, we know from past work that nutrient flushes right after a prolonged dry period are typical, which in turn could be a significant driver of higher CO₂ from both heterotrophic and autotrophic sources.

Again, a nice study. If the above issues could be given some greater clarity, then I think it would merit publication.

Reviewer #3 (Remarks to the Author):

NCOMMS-17-07841-T

Author: O'Connell CS

Title: Drought drives rapid shifts in tropical rainforest soil biogeochemistry and greenhouse gas emissions

General Comments

The study takes advantage of a natural experiment, drought, to examine tropical forest soil biogeochemistry and greenhouse gas dynamics. This is an important area of research that is appropriate for publication in the journal. The study is done well, and the data are interesting. The paper is well written.

I have only a few minor comments and concerns.

Specific Comments

- 1) The abstract reads well.
- 2) The logic presented in the introduction is sound.
- 3) The results are presented well.
- 4) The pre-drought rates of CO₂ emission seem, to me, to be a bit low. These sites are not large sources of CO₂. Do these low rates, in part, determine response to drought? In other words, sites with high pre-drought rates experience declines with drought, whereas low pre-drought rates will increase?
- 5) The conclusion on page 9 is okay but a bit flat, i.e., biogeochemistry is complicated but important. Perhaps you could provide a bit stronger conclusion that is quantitative. Consider using the percentage changes in CO₂ equivalents (Page 8, line 170) as the conclusion.
- 6) All of the tables and figures are necessary.
- 7) The statistical analysis is appropriate, by my standards.

Technical Comment

- 1) Page 2, line 28: pseudo-continuous. That is a new one to me. I assume that you mean continuous but with gaps in the data.
- 2) Page 4, line 74: The sentence is a bit confusing because the meaning of 'decline' is uncertain. Do you mean decline to the lowest level, or decline as the largest departure from normal?
- 3) Page 8, line 169: CH₄ emission that offsets CH₄ consumption is a good finding. You say it well, but you might mess around a bit with the wording so that no one misunderstands the findings. These soils are most sources of atmospheric CH₄, not sinks.
- 4) Page 12: I read the gas flux section several times. It reads well. However, the frequency of sampling a given site is not clear to me. Perhaps I missed the sampling schedule. Did you sample, on average, every day? Once a week? Whenever? You might try to make this clearer (?).

Point-by-point response to the referees' comments

**Reviewer #1:**

***1. Summary: This article describes the first example of “continuous-monitoring” automated***
***chamber measurements of co2 and methane emissions from a tropical forest. Over one year,***
***soil emissions were measured over a total of 150 unique days at three topographic positions in***
***a lowland tropical rainforest. The study period encompassed a significant drought, enabling***
***description of pre- during- and post-drought emissions, which varied. Soil o2 and h2o contents***
***were simultaneously monitored through time in the same plot, as were soil inorganic and***
***available P contents.***

We thank Reviewer #1 for comments that allowed us to substantially improve the manuscript.

***2. Temperature data are notably absent.***

We now include these data. See our response to comment #5 below.

***3. Data are presented in figures only, minimizing their usefulness to others.***

We now include Tables S3 and S4 which present the data in Figures 3 and 4 as tables so that
readers can easily look up values. We also include Table S5 so that results from each flux
chamber are reported from each drought-related period (see below). Data will be made available
through the LUQ-LTER and LCZO data management portals concurrently with paper release
date.

***4. The main findings are (I) drought increases co2 emissions but decreases ch4 emissions (Fig.***
***S5), possibly via O2, and (II) the responses of both co2 and ch4 emissions through time varied***
***among topographic positions (Fig. 3).***

Yes.

***5. The lack of temperature data precludes the conclusion that dry soils, which cause soils to***
***warm, caused the observed changes in gas fluxes.***

The reviewer's point is well taken; we originally did not show these data as temperature does not
vary much in humid tropical forests. We now include soil temperature data from the same sensor
array (Figure S6) and air temperature data from a nearby weather station (Figure S6). We
compare soil and air temperature patterns and soil temperature and GHG patterns (Figure S7) and
discuss these results in a new paragraph in the discussion section (lines 224-235).

***6. Line :***

***25 “key nutrients” includes only two forms of P, organic and inorganic, or are you including***
***Fe redox state as a nutrient? Would “soil phosphorus” be more appropriate?***

Changed. We also include a clause for the reader emphasizing P's importance in this system (line
30).

31 It is important that the reader understands what “landscape-scale” means. According to the article title, it means a tropical rainforest. On the other hand, a standard hectare is referred to in the methods. In my experience, no single hillslope accurately reflects a tropical rainforest. I recommend that you avoid the “landscape” concept and word. This was a case study of a single hillslope; treat it as such.

We now use the terms “topography” or “catena” when referring to the replicate transects of sensor and chamber data. When we scale up from our results, we now refer to “ecosystem” effects.

48 I suspect that “there are no continuous” underlies a perceived value of this research. The trace gas measurements reflect repeated measures from nine distinct chambers. How that sample size was magnified by segregating by position and time-period needs to be explained more clearly, for the reader to properly interpret the findings. As I read it, there should be 36 total numbers, with a sample size and error terms, which underlie all of bars in Figure 3. As those are that backbone of this work, I suggest that they be included in a table in the appendix.

We added two sentences to the introduction (lines 52-58) to make this framing clearer and cite literature calling for the need for these types of data (Chambers 2012, line 58).

The reviewer is correct regarding the replicate chambers and how they were distributed in space and time when used for Fig. 3, but the sample was in no way “magnified”. We identified and separately explored different periods within the record that were identified via statistical approaches (lines 406-416). Our design was conceived to also allow us to parse out the effect of topography. We added text to the methods experimental design section (beginning line 296; highlighted text) that clarifies how the chambers were located.

We have included a table in the supplement with the GHG/chamber/time period values mentioned (Table S5, which includes more information than what is shown in Fig. 3, and Table S3 which is the table version of the data represented in Fig. 3).

51-56 There are no objectives, hypotheses or questions stated, and thus no purpose identified, for this article. As a result, there is no logical organization, and the reader is forced to simply read along and see what comes next. Compounding this difficulty, there are no subheadings or other clues that might provide organization, and guide the flow of logic for the reader. Then, the concluding and introductory paragraphs are nearly the same. Your interesting findings, if they exist, are well disguised within the disorganization.

We now include the key questions and hypotheses addressed in this study, which further clarifies the purpose of the article (lines 65-76). We have added headings following the Nature Communications format.

82-84 The clear statements of findings are welcome. However, there is no reference to any interaction term, so the statistical model is not obvious. Compare the (overly tiny) statistical results shown on Figure 3 with those on Figure 4: they are different! This suggests selective statistics, and I am quite sure that you want to be certain to avoid such an appearance. Basically, as a result, none of the results that you present are clearly derived from discernible statistical tests. This problem is magnified by the complete lack of any information about degrees of freedom or sample sizes, that is, basic information that informs the reader that the

155 *data was properly analyzed and interpreted. It is not included in the methods or statistical*
*methods either.*

We have increased the font for the statistics information in Figures 3 and 4 so they are no longer
“overly tiny”. We used repeated measures for the GHG data as these were true repeated measures
(same chambers and same locations sampled over multiple time points). We also added a text
line in the place of Figure 4’s “interaction term” on Figure 3 clarifying for the reader that we used
a repeated measures term in our model. We used a regular ANOVA for the soil variable data
(e.g., soil P, Fe, pH) as these were not true repeated measures (sampled twice from randomly
located sites within topographic zones). We also added a table (S6) giving details of the
statistical tests used (including sample sizes, degrees of freedom, initial test results and posthoc
test results).

We explored rerunning the greenhouse gas statistics using an interaction term. We ran the tests
and this did not change the overall results. It is important to note that these tests can lead to
covariate interactions that make the default contrasts inappropriate. That is, the repeated
measures error term can function, in effect, as the interaction term for a model, leading to the
inappropriate possibility of accounting for interactions twice. In this case, the repeated measures
error term (repeated Date by Drought Period and Topographic Location) incorporates the
interaction between Drought Period and Topographic Location directly. Thus, we chose to
maintain our original statistical approach.

We have added some clarifying text to the statistical methods section (lines 417-438) and pointed
readers to the new tables.

***97 The abstract (line 26) highlights only that inorganic P declined, without mentioning the***
***opposing and apparently equal increase in organic P, and this same bias occurs in the***
***concluding paragraph, at line 178. This again appears to be selective presentation, which is***
***misleading.***

We now present inorganic and organic P results in the abstract (line 27). We also included
increases in organic P in the list of indirect ecological impacts of drought in the conclusion
paragraph (line 273). Note that both results were originally presented in the text.

***123 The interpretation of the soil moisture -- O2 relationship via linear regression is not even a***
***tiny bit convincing. For one thing, sample sizes strongly influence R2 and slopes. For another,***
***it is obviously not a linear relationship. The conclusion made seems unwarranted, and are***
***poorly described (in terms of what it is and what change occurred). Delete***

We agree that this is not central to the main message, thus we deleted both the written results and
the supplemental figure.

***157 It is gratifying to see that you did not forget to mention hot spots and hot moments. But it is***
***unfair to ask the reader to first find Figure S3, and then interpret it properly, and then come to***
***the same conclusion that you did, i.e., “a marked drop”. Figure S3 presents means and***
***standard deviations only – no hot spots or moments. Also, it is not clear how hot spots emerged***
***when there were only nine chambers in (apparently) fixed positions. Are hot spots and hot***
***moments the same things?***

We left the reference to Figure S3, but added sentences to the text that highlight the difference in
how large the large fluxes were for each time period (lines 175-180). We report the 90th

percentile fluxes during each drought period as a measure for whether or not large fluxes were as
prevalent through time. Specifically, we added the following: “The 90th percentile of pre- and
post-drought emissions of CH₄ were an order of magnitude higher than the 90th percentile flux
rate during the drought and drought recovery periods (90th percentile rates of 11.77, 1.88, 1.66,
22.01 nmol m⁻² s⁻¹ during each consecutive drought period).” We dropped the reference to hot
spots as we primarily refer to hot moments in this instance (although we do mention the specific
chamber locations, which can be interpreted as “spots”).

***Figure S5 This figure would be far more clear if (a) it included the entire period of study, with***
***four time periods and (b) used molar values on the vertical axis. The values used herein are***
***very challenging but could be simple. Simply use CO₂-Ce (mol area-1 time-1). But state your***
***correction for methane (e.g., 86?), which you do not, anywhere.***

We included vertical lines on Figure S5 indicating the breaks between drought time periods. We
keep the convention of reporting CO₂e as a mass (kg or Mg ha⁻¹) as this makes the data more
comparable to the rest of the literature. All the data will be available for any who choose to
compare with less conventional units.

We had previously stated our methane correction in the methods section (line 439) and it is now
in the main text (line 186).

***173-175 Thresholds and lags were also observed. Again, however, these are presented as***
***results without clear justification, and free of any statistical basis or meaning.***

We felt that this was obvious from marked decline in moisture and similarly rapid decline
increase in O₂ relative the rest of the record shown in the figures. In order to avoid confusing
terminology, we edited the sentence in question so that it no longer uses the terms “threshold” or
“lag,” now reading “Across the observed topographic gradient, the effects of drought
belowground occurred suddenly, with large changes in moisture and O₂ events over the one- to
three-week time scale, and resolved slowly (over 12 weeks), both extending the effective drought
period and decoupling soil moisture and soil redox dynamics” (line 192, highlighted portion is
new clarifying text).

***210 Figure S2 lacks important information that also is not provided in the text methods: What***
***is the elevation difference between the ridgetop and valley? What were distances between***
***H₂O/O₂ transects? What was the overall plot size?***

We have included elevation distance, slope angle, distances between transects and overall plot
size in the experimental design methods section (line 298-303). We’ve also amended Figure S2.

***332 The data from the single hillside array are applied to the forest landscape as a whole, to***
***deduce the “rainforest” response to drought. Is this warranted?***

We have replaced the word “landscape” with the term “ecosystem”. The five instrumented
transects incorporate considerable variability, and were chosen to be representative of the
topographic variability in these ecosystems (see line 311 for a new sentence). Our scaling
exercise is a first approximation, and is labeled as such (“first approximation of fluxes at the
ecosystem scale”, line 184).

***335 This statistical approach presumes that the observed fluxes were normally distributed and***
***had similar variances among groups, including across time periods when moisture content***

*varied. This is exceedingly unlikely. Normal statistics are unwarranted unless you document*
*that variances among compared groups were not different.*

Though traditionally in biogeochemistry we don't expect to see trace gas fluxes represented well
using parametric models, our dataset is in the uncommon position of having a large enough
sample size that, when we ran ANOVA diagnostics, the key assumptions of ANOVA were not
violated. We log transformed the CO₂ flux data in the original submission of this manuscript to
better meet the assumption of normality, and kept that transformation in place here. CH₄ fluxes
did not require transformation. We have included in this revision two supplemental figures with
diagnostic information, Figure S8 and S9 (CO₂ and CH₄, respectively). We include plots
demonstrating the normality of model residuals (QQ plots and residual histograms), boxplots
illustrating the distribution of residuals across groups (as these are two-way ANOVAs, we
include boxplots that show residuals grouped by drought time period and by topographic
location), plots investigating homoscedasticity (a spread-location plot) and a trellis plot. We also
include the histograms of the dependent variables (after transformation, in the case of CO₂) for
the reader's reference.

In parametric statistical tests such as ANOVA a key assumption is that conditional, not marginal,
distributions must be normal, i.e., the model residuals must be normally distributed but the
dependent variable observations need not be (see Dalgaard 2008). This assumption can be tested
by inspecting the QQ plots and residual histograms in Figure S8 and S9.

The reviewer's point that variances across compared groups ought not to differ is well taken.
This can be checked by inspecting the two boxplots of residuals across groups in Figure S7 and
S8. Here valley CO₂ fluxes have lower spread than the other two topographic regions and pre-
drought CH₄ fluxes have lower spread than the other drought time periods. However,
determining how much difference across groups is "too much" is often a judgement call, and we
did not believe that either of these patterns was disqualifying.

We note that repeated measures ANOVAs are commonly used in biogeochemistry and generally
preferred for a number of reasons in comparison to the non-parametric options available to us. A
Friedman Test can be used as a non-parametric equivalent of a one-way repeated measures
ANOVA and a Scheirer-Ray-Hare test (a spinoff of a Kruskal-Wallis test) is roughly a non-
parametric two-way ANOVA, but isn't compatible with repeated measures and requires equal
sample sizes between groups, while our data has uneven sampling across time. Neither test, nor
any other that we found, suited our needs for an alternative to a two-way repeated measures
approach. Since we (a) felt confident that our original statistical approach was defensible based
on the diagnostics outlined above, and (b) felt that the available non-parametric statistical options
were both opaque and imperfect, we retain our original statistical methods in the revised version
of the paper.

Citation: Dalgaard, P. Introductory Statistics with R (2008).

***Reviewer #2 (Remarks to the Author):***

*This is an interesting and solid study that adds some unique information to a growing*
*understanding of drought in tropical forests. The substantial amount of data spanning the*
*drought and post-drought period offers a real plus, and in general I think this MS is worth*
*considering for publication in Nature Comms. I do have a couple comments I believe need*
*further consideration:*

We thank Reviewer #2 for comments that allowed us to substantially improve the manuscript.

***- The organic P increases are quite large – surprisingly so. They seem too large to be explained***
***by a difference in decomposition, and given the increase in CO₂ emissions invoking a decrease***
***in decomp for the organic P change doesn't really make sense. The leaf flush perhaps makes***
***more sense but even that seems a bit tenuous given the time dynamics. I don't have a great***
***explanation for what truly drove this, but would like to see the authors think on this one some***
***more. E.g. are you seeing a big increase in microbial uptake of inorganic P pools under***
***drought and conversion of that Pi into organic pools? Not sure...***

We improved our discussion of soil P changes by offering several potential explanations for the
organic P increases in a new discussion paragraph (lines 244-260). We add to the paper two new
potential mechanisms that hinge on the microbial community: “Drought conditions could have
led to a slowdown of extracellular phosphatase activity, an effect that has been observed in other
systems^{49,50}. This pattern could reconcile higher CO₂ emissions with higher organic P
concentrations, as microbial decomposition could proceed without liberating the P in soil organic
matter. Alternatively, rapid and considerable microbial uptake of the newly available inorganic P
would have been measured as organic P” (lines 250-255).

***- One of the truly hard things to do in interpreting the drought response of CO₂ is to know how***
***much is from heterotrophs vs tree dynamics (so I'm sympathetic). There is some speculation in***
***the MS about reasons for the change, but not much in terms of any supporting information***
***that might help point the way on what's most responsible. Do you have any direct evidence for***
***changes in decomposition rates and/or changes in plant C allocation? E.g. note the Doughty et***
***al Nature paper on drought response that focused on C allocation responses in trees,***
***demonstrating how critical this can be to interpreting the true C balance response of the***
***system.***

We agree that a subtler treatment of the observed CO₂ flux patterns was needed. We've now
added several paragraphs that flesh out some potential explanations for observed changes in GHG
fluxes. We discuss CO₂ in two paragraphs (lines 202-235) and CH₄ in a paragraph that extends
the previous manuscript's shorter explanation of flux shifts (lines 236-243).

***- Likewise, the post-drought further increase in CO₂ could be coming from multiple sources,***
***again would be good to know if there are any supporting data to help interpret. Also, we know***
***from past work that nutrient flushes right after a prolonged dry period are typical, which in***
***turn could be a significant driver of higher CO₂ from both heterotrophic and autotrophic***
***sources.***

We agree here as well that an improved discussion was warranted. We specifically reference
these two ideas in the CO₂ paragraph found in lines 202-223.

***Again, a nice study. If the above issues could be given some greater clarity, then I think it***
***would merit publication.***

Thank you for your helpful feedback.

***Reviewer #3 (Remarks to the Author):***

***NCOMMS-17-07841-T***

*Author: O'Connell CS*

*Title: Drought drives rapid shifts in tropical rainforest soil biogeochemistry and greenhouse*
*gas emissions*

*General Comments*

*The study takes advantage of a natural experiment, drought, to examine tropical forest soil*
*biogeochemistry and greenhouse gas dynamics. This is an important area of research that is*
*appropriate for publication in the journal. The study is done well, and the data are interesting.*
*The paper is well written.*

*I have only a few minor comments and concerns.*

We thank Reviewer #3 for comments that allowed us to substantially improve the manuscript.

*Specific Comments*

*1) The abstract reads well.*

*2) The logic presented in the introduction is sound.*

*3) The results are presented well.*

*4) The pre-drought rates of CO₂ emission seem, to me, to be a bit low. These sites are not large*
*sources of CO₂. Do these low rates, in part, determine response to drought? In other words,*
*sites with high pre-drought rates experience declines with drought, whereas low pre-drought*
*rates will increase?*

The reviewer raises an interesting question, that unfortunately cannot be answered from these
data on a single drought event in one ecosystem. If more data on drought become available from a
wider range of tropical forests, it will be interesting to see if background conditions can dictate
the sign of the response.

*5) The conclusion on page 9 is okay but a bit flat, i.e., biogeochemistry is complicated but*
*important. Perhaps you could provide a bit stronger conclusion that is quantitative. Consider*
*using the percentage changes in CO₂ equivalents (Page 8, line 170) as the conclusion.*

We agree that the final paragraph acted too much like a summary and have rewritten it (lines 261-
275). We now emphasize the change in CO₂e as suggested.

*6) All of the tables and figures are necessary.*

*7) The statistical analysis is appropriate, by my standards.*

*Technical Comment*

*1) Page 2, line 28: pseudo-continuous. That is a new one to me. I assume that you mean*
*continuous but with gaps in the data.*

After learning that this term presented some confusion, we replaced it with “high frequency”
(e.g., high-frequency data) for ease of understanding. Replacements were made on line 30 and
69.

**2) Page 4, line 74: The sentence is a bit confusing because the meaning of ‘decline’ is**
**uncertain. Do you mean decline to the lowest level, or decline as the largest departure from**
**normal?**

We meant the latter (decline as the largest departure from normal) and have edited the text to
reflect this (line 100-102).

**3) Page 8, line 169: CH₄ emission that offsets CH₄ consumption is a good finding. You say it**
**well, but you might mess around a bit with the wording so that no one misunderstands the**
**findings. These soils are most sources of atmospheric CH₄, not sinks.**

We have revised the sentence for clarity and accuracy (current version of this sentence can be
found on line 181-183).

**4) Page 12: I read the gas flux section several times. It reads well. However, the frequency of**
**sampling a given site is not clear to me. Perhaps I missed the sampling schedule. Did you**
**sample, on average, every day? Once a week? Whenever? You might try to make this clearer**
**(?).**

We added a paragraph to the methods section (lines 346-359) that clarifies our ideal gas sampling
schedule (daily, 2-hour cycles running continuously looping through each automated chamber)
and the constraints on continuous sampling that sometimes altered that idealized schedule
(instrument malfunctions or fallen debris preventing chamber closure).

REVIEWERS' COMMENTS:

Reviewer #2 (Remarks to the Author):

I appreciate the thorough and thoughtful replies from the authors to the prior concerns raised. I have no additional concerns and believe this MS warrants consideration for publication at this stage.

Reviewer #3 (Remarks to the Author):

I reviewed the original version of the paper. Thank you for addressing my comments and concerns.

**Point-by-point response to the referees' comments**

**Reviewer #2 (Remarks to the Author):**

*I appreciate the thorough and thoughtful replies from the authors to the prior concerns raised.*
*I have no additional concerns and believe this MS warrants consideration for publication at*
*this stage.*

We thank Reviewer #2 for comments that allowed us to substantially improve the manuscript and
are pleased that they believe they were appropriately addressed.

**Reviewer #3 (Remarks to the Author):**

*I reviewed the original version of the paper. Thank you for addressing my comments and*
*concerns.*

We thank Reviewer #3 for comments that allowed us to substantially improve the manuscript and
are pleased that they believe they were appropriately addressed.
